# A Proteomics Insight into Advancements in the Rice–Microbe Interaction

**DOI:** 10.3390/plants12051079

**Published:** 2023-02-28

**Authors:** Lirong Wei, Dacheng Wang, Ravi Gupta, Sun Tae Kim, Yiming Wang

**Affiliations:** 1Key Laboratory of Integrated Management of Crop Disease and Pests, Ministry of Education, Department of Plant Pathology, Nanjing Agricultural University, Nanjing 210095, China; 2College of General Education, Kookmin University, Seoul 02707, Republic of Korea; 3Department of Plant Bioscience, Pusan National University, Miryang 50463, Republic of Korea

**Keywords:** rice, proteome, immunity, plant–pathogen interactions

## Abstract

Rice is one of the most-consumed foods worldwide. However, the productivity and quality of rice grains are severely constrained by pathogenic microbes. Over the last few decades, proteomics tools have been applied to investigate the protein level changes during rice–microbe interactions, leading to the identification of several proteins involved in disease resistance. Plants have developed a multi-layered immune system to suppress the invasion and infection of pathogens. Therefore, targeting the proteins and pathways associated with the host’s innate immune response is an efficient strategy for developing stress-resistant crops. In this review, we discuss the progress made thus far with respect to rice–microbe interactions from side views of the proteome. Genetic evidence associated with pathogen-resistance-related proteins is also presented, and challenges and future perspectives are highlighted in order to understand the complexity of rice–microbe interactions and to develop disease-resistant crops in the future.

## 1. Introduction

Pathogens are a major threat to crop production. Therefore, the use of chemical pesticides has greatly expanded in recent decades. Although pesticides increase crop productivity by limiting infection, the extensive and continuous use of pesticides can be a threat to the environment due to their biomagnification and persistent nature. Therefore, the development of effective and environmentally friendly strategies for crop disease control is an important prospect for plant scientists and breeders.

Systemic investigations of plant–microbe interactions require high-throughput techniques such as genomic sequencing, transcriptome analysis, and proteome and metabolome profiling. Proteomics was initially used as a tool for the identification of all proteins; however, this state-of-the-art approach has been extensively utilized in the last two decades for the determination of diverse protein properties, such as their amino acid sequence, relative and absolute abundance, post-translational modifications (PTMs), and protein–protein interactions [1,2]. Initially, Biemann and colleagues utilized peptide sequencing by mass spectrometry (MS) in 1966. The terms proteome/proteomics appeared in print in 1995, three decades later [3,4]. The development of the two-dimensional gel electrophoresis (2-DGE) technique, coupled with MS identification, was a foundational stone in proteomics which facilitated the comparison of protein profiles among different samples through the on-gel quantification of protein spots [5]. Subsequently, label-based protein quantification strategies, such as two-dimensional fluorescence difference in gel electrophoresis (DIGE) [6], isotope-coded affinity tags (ICATs) [7], isobaric tags for relative and absolute quantitation (iTRAQs) [8], tandem-mass tags (TMTs) [9], and stable isotope labeling by/with amino acids in cell culture (SILAC) [10], were developed that fostered an even deeper protein profiling of biological samples. Recently, label-free quantitative proteomic approaches have been successfully applied in genome-wide proteomics investigations. For instance, the LC-MS/MS-based identification covers almost 70% of the proteins expressed in humans [11,12], whereas a coverage of approximately 50% was achieved by Arabidopsis [13]. These studies provided insights into the changes in protein abundance, post-translational modifications, and protein–protein interactions in a more systematic way. Additionally, these methods have been frequently utilized to investigate relative large-scale protein abundance alternations in plants during their interactions with different microbes [14,15].

The development of protein separation and identification techniques significantly improved the protein detection sensitivity in plants. However, other factors, such as sample purity and the coverage of protein databases, are still major bottlenecks in the proteomics analysis of plant–microbe interactions [14,15,16]. The successful infection of pathogens is the first and critical step for the induction/alternation of the abundance of defense-related proteins during host–microbe interactions. The well-infected plant tissues need to be collected when infection symptoms appear, rapidly frozen in liquid nitrogen, and ground into a fine powder for protein extraction (Figure 1). Moreover, protein fractionation based on different cell organelles or PTMs significantly improves the detection of low-abundance proteins and protein PTMs (Figure 1) [17,18]. Therefore, the selection of protein separation and enrichment methods would also strongly affect the throughput level, sensitivity, and degree of accuracy during proteomics [19]. Between gel-based and gel-free proteomics approaches, the former generally exhibits a high degree of accuracy, whereas the latter approach shows a high level of throughput and sensitivity [20]. Additionally, advancements in bioinformatics, including the improvement of protein databases, algorithms for MS identification, and protein annotation, further benefit and further provide a deeper understanding of biological samples [21].

During their long-term co-evolution with pathogenic microbes, plants developed a multi-layered immune system [22]. The recognition of pathogen-associated molecular patterns through membrane-localized pattern recognition receptors activates a rapid and transient defense response which is commonly known as a pathogen-associated molecular pattern (PAMP)-triggered immunity (PTI) [22,23]. To suppress the PTI, pathogens deliver effector proteins into the host cells through different protein secretion mechanisms that increase their susceptibility, a process that has been named “effector-triggered susceptibility” (ETS) [22]. In response to ETS, the cytoplasmic, localized, nucleotide-binding leucine-rich repeat receptors of the host recognize pathogen-derived effectors directly or indirectly and display a remarkable robust immunity termed “effector-triggered immunity” (ETI) [24]. Recent research has shown that both PTI and ETI are interlinked, and PTI is required for the induction of ETI, as reported in Arabidopsis [25]. Moreover, the activation of PTI enhances the hypersensitive response during ETI [26], suggesting that the activation of both PTI and ETI is required for complete resistance against pathogens. Systemic, acquired resistance is another layer of plant immune response in which pathogen-challenged plants establish an immune memory in both local and uninfected distal leaves that can mount a more effective immune response during a secondary infection to enhance resistance against a broader range of pathogens [27].

A group of pattern recognition receptors (PRRs) involved in the identification of PAMPs and resistance genes involved in the identification of cytoplasmic effectors have been identified in rice using genomics, transcriptomics, and proteomics approaches. The PRRs identified in rice so far include CEBiPs, LYPs, OsFLS2, and XA21, which are involved in the recognition of PAMPs, including chitin, PGN, flagellin, and RaxX21, respectively, to trigger PTI responses [28,29,30,31,32]. The rice NB-LRR proteins, including Pib, Pita, Piz-t, Pigm, and Xa1 are involved in the activation of ETI through the recognition of pathogen-secreted effectors in the host cytoplasm [33]. In addition, the downstream signaling components identified in rice so far include the small G-protein, MAPKs (OsMAPK3/OsMAPK6), and various transcription factors such as OsWRKYs, OsMADSs, and OsNACs. The characterization of genes that are required for host immunity provides novel clues for breeders to generate crops with a broad-spectrum resistance against bacterial and fungal pathogens [34,35,36,37]. For instance, the inclusion of pathogen-responsive upstream open reading frames (uORFsTBF1)-mediated translational control of AtNPR1 exhibited a broad-spectrum disease resistance without compromising plant fitness in rice [38]. Likewaise, Pigm, a resistance gene from the rice variety Gumei 4, was characterized as a broad-spectrum resistant gene. Its epigenetic modification is known to be required for the Pigm-mediated broad resistance against rice blast fungus [36]. These findings suggest that rice proteins can be targeted to generate rice cultivars with an improved pathogen resistance. This is a more cost-effective, highly efficient, and environmentally friendly strategy than the use of chemical pesticides. Therefore, a deeper understanding of the plant immune system would provide novel clues and benefit the design and engineering process of resistant crops.

In recent decades, a number of proteomics studies have been carried out to investigate intercellular responses, metabolic and signaling pathways, and key regulators involved in rice–pathogen interactions. In this review, we summarize the recent progress of proteome identifications, post-translational modifications (PTM), and the functional validation of identified proteins in rice–microbe interactions. Moreover, future perspectives are also provided to illustrate the trends in proteomics applications in plants and to provide clues for researchers and breeders to generate highly resistant crops in the future.

## 2. Interactions between Rice and Pathogenic Bacteria

Bacteria have evolved a wide range of strategies to invade and colonize host plants. The commensal bacteria, which are generally non-pathogenic, adhere to and multiply on the cell surface, while the pathogenic bacteria enter host tissues through openings such as stomata and wounded regions and proliferate in the apoplastic region where the first interaction between the host and pathogen-derived proteins takes place [39]. Therefore, proteomic tools have been utilized to understand the host protein alternations upon interaction with commensal and pathogenic bacteria (Table 1).

*Xanthomonas oryzae* pv. *oryzae* (*Xoo*), the agent that causes rice blight, is one of the most devastating diseases of rice. Several research groups utilized 2-DGE-based proteomic tools to identify the proteins involved in *Xoo* resistance in rice. This research was recently been reviewed [1]. Due to technical limitations, only a few proteins involved in *Xoo* resistance have been identified to date. Defense-related proteins such as PR5, PBZ1, SOD, and peroxiredoxin were shown to be highly accumulated in the rice plant after *Xoo* inoculation [40]. In the plasma membrane (PM) fractions, PM-associated H^+^-ATPase, protein phosphatase, hypersensitive-induced response protein, and prohibitin were found to be significantly induced upon *Xoo* infection, suggesting the possible roles of these proteins in resistance to rice blight [41]. Among those, prohibitin is of particular interest as it was also identified in mammalian cells exhibiting antiproliferative functions [68]. In animals and yeasts, prohibitin was reported as a pleiotropic protein related to metabolism, senescence, and immunity [69]. *Arabidopsis thaliana* and *Nicotiana tabacum* also contain a functional prohibitin, suggesting that prohibitin may exhibit a similar function in plants, yeast, and mammals [70]. In plants, prohibitin protein is generally localized in the mitochondria and chloroplast [70,71,72]. A recent report showed that the silencing of the prohibitin gene NbPHB2 caused severe growth inhibition, leaf yellowing, and cell death, and mutant tobacco plants were hypersensitive to oxidative stress [72]. In rice, the prohibitin gene NAL8 participates in the leaf and spikelet development by modulating the mitochondrial and chloroplast stability, suggesting a crucial role of prohibitin in plant growth [71]. Subsequent reports showed that PROHIBITIN3 (PHB3) forms a complex with the salicylic acid (SA) biosynthesis gene ICS1 and contributes to a SA-mediated stress response in Arabidopsis [70].

More recently, Zhang and colleagues performed a TMT-based quantitative proteome analysis to investigate the protein changes during incompatible and compatible interactions between *Xoo* and rice [73]. Secondary metabolites including phenylalanine, flavonoids, phenylpropanoids, phenolic phytoalexins, and phytohormone SA were highly accumulated under incompatible interactions, suggesting secondary metabolites may directly suppress the infection of bacterial pathogens in rice. However, since limited experimental evidence has been provided to date, the exact role of plant-derived secondary metabolites in bacterial interactions is still elusive. Calcium signaling and MAPK signaling cascades, including CDPK13, OsMKK4, and OsMPK6, were significantly induced. This is consistent with a previous finding that OsMAPK6, which functions downstream of OsMAPKK10.4-OsMKK4, was required for the resistance against *X. oryzae* pv. *oryzicola* (*Xoc*) by activating SA and jasmonic acid (JA)/ethylene (ET) signaling in rice [47,74]. Moreover, OsMKK4 is also required for the regulation of phytoalexin biosynthesis in rice during *M. oryzae* infection [75]. Interestingly, the OsMAPKKK10-OsMAPKK4-OsMAPK6 cascade is also involved in the regulation of rice grain size and panicle development [76,77]. Therefore, the modification of OsMKK4-OsMAPK6 signaling in rice may have a high potential in generating rice cultivars with a high resistance and high yield.

Wild rice, *O. meyeriana*, exhibits a higher resistance to *Xoo* compared with japonica rice varieties; therefore, stable somatic hybrid lines of *O. meyariana* and *japonica* rice cultivars were generated. These hybrids exhibited a broad resistance to *Xoo* strains isolated from the Philippines and China [39]. Wild rice enhances *Xoo* resistance by increasing the accumulation of proteins related to photosynthesis, metabolism, ROS metabolism, and defense [42]. However, the molecular mechanism is still elusive. Utilization of the wild rice and japonica hybrid population to identify key immune regulators would help to develop rice with a broad resistance.

A recent report focused on the identification and profiling of low-abundant proteins during compatible and incompatible interactions between rice and *Xoo* [78]. A protamine-sulfate-based method was employed to enrich the low-abundance proteins that were subsequently identified and quantified by a label-free quantitative proteomics approach [78]. Compared with the compatible interactions, the high accumulation of protein kinases, such as calcium-dependent protein kinases, PTI1-like tyrosine-protein kinase 1, protein kinase domain-containing protein, and serine/threonine-protein kinase, under an incompatible interaction indicates that signal transduction through phosphorylation by protein kinases is required for the ignition of immunity in rice. Interestingly, a mitochondrial arginase-1 (OsArg1) also exhibited a high abundance under incompatible *Xoo* interaction conditions. The overexpression of OsArg1 significantly enhanced rice resistance against *Xoo* and enhanced the expression of defense-related genes Chitinase II, Glucanase I, and PR1, suggesting the involvement of OsArg1 in *Xoo* resistance [48]. This result suggests the possible role of mitochondria in pathogen resistance. Therefore, the application of mitochondria fractionation and proteomics identification would benefit the understanding of mitochondrial responses in pathogen resistance. Moreover, the upstream PRRs and downstream transcription factors, which are required for the protein-kinase-mediated signaling, have not been well identified. These results strongly motivate us to establish an intact signaling network in rice in response to *Xoo* infection. Therefore, a phospho-proteomics analysis on the plasma membrane and nuclear fractionations during pathogen infection need to be performed in the future.

The apoplast is the first region in which *Xoo* proliferates after invasion. It is also the first place where host–microbe interactions take place, including host recognition and bacterial growth suppression. Therefore, efforts have been made to investigate the changes in the apoplast proteome/secretome upon pathogen challenge [58,79]. First, a 2-DGE/DIGE-based secretome analysis was performed in the suspension-cultured callus of *O. sativa* and *O. meyeriana* challenged by *Xoo.* This led to the identification of seven and thirty-four differentially modulated proteins, respectively [43,44]. These differentially accumulated proteins were related to energy production, protein metabolism, and defense (mainly endo-1,3-beta-glucosidases, GH16 family proteins, and stress-responsive proteins), redox states (mainly peroxidases), and cell wall modifications (mainly expansions). Furthermore, peroxidase activity was much more significantly induced in *O. meyeriana* than in the susceptible race of *O. sativa*. Subsequently, a label-free quantitative proteome analysis was employed to understand the interaction between rice–*Xoo*, which provided a global view of their interaction for the first time [45]. A total of 727 and 186 proteins secreted from *Xoo* and rice, respectively, were identified. Of these proteins, proteins related to oxidative stress (twenty-one peroxidase isoforms, five peroxiredoxin isoforms, three Cu/Zn-SOD isoforms, two ferredoxin isoforms, and one glutathione S-transferase, thioredoxin, and ascorbate peroxidase) were highly induced in rice, indicating the activation of the antioxidant defense system to detoxify the stress-induced ROSs. These findings are consistent with a recently published report which showed that the resistance of rice to disease1 (ROD1) (SNP1A) exhibits broad-spectrum disease resistance without an obvious reduction in yield through modulating ROS balances in rice [37]. Other proteins accumulated in rice upon *Xoo* infection include those related to carbohydrate metabolism, proteolysis, ion transport, and defense (chitinase, GH17 family proteins, thaumatin-like proteins, and pathogenesis-related bet VI family protein) [45].

Phosphorylation is one of the most important post-transcriptional modifications (PTM) during a host–microbe interaction. It plays a central role in signal transduction. A quantitative phosphoproteome analysis was carried out to investigate the PTM changes in resistance and susceptible rice lines during *Xoo* inoculation [46]. Phosphopeptides were enriched by TiO2-MOAC (metal oxide affinity chromatography) and identified by nLC-MS/MS. This led to the identification of more than 2000 phosphopeptides, representing 1334 and 1279 unique proteins at 0 h and 24 h post-infection with *Xoo*. Several conserved phosphorylation motifs were identified among the identified phosphosites, including the [RxxS] recognized by CaMK and MAPKK, [SP] recognized by MAPK, [Rxxs] by SnRK and CDPK, and [SxS] by receptor kinases. Most of the differential phosphoproteins were localized in the nucleus, where the phosphorylation of transcription factors takes place. However, the peptide intensity relating to transcription regulation was not affected, suggesting that these are mainly regulated post-translationally. Interestingly, the phosphorylation of two PP2Cs (OsPP2C27 and OsPP2C57), which are the negative regulators of ABA signaling, was increased, suggesting a negative role of ABA in *Xoo* resistance in rice [80]. This observation was further confirmed by the work of Liu and co-workers, in which it was observed that the reduction of ABA content through the over-expression of NAC transcription factor ONAC066 led to an enhanced resistance against *Xoo* in transgenic rice [81].

*Xoc* causes narrow, dark-greenish, water-soaked, interveinal streaks of various lengths known as rice leaf streak. In order to understand rice responses to *Xoc* infection, protein changes in indica cultivar 9311 were analyzed through 2-DGE MS [49]. Among 1500 protein spots, 32 upregulated proteins were identified that were related to cell metabolism and disease resistance such as pathogenesis-related proteins. In addition, several putative receptor kinases were also identified, suggesting the involvement of those proteins in rice defense signaling. Interestingly, the abundance of OsMAPK6 was significantly increased, suggesting a possible role of OsMAPK6 in *Xoc* resistance together with *Xoo* resistance.

Taken together, these proteomics studies illustrate that after sensing bacterial signals, rice triggers a rapid immune response through the phosphorylation of CDPKs, CaMK, and MAPKs (Table 1). These phosphorylation events further induce the phosphorylation of transcription factors, including OsWRKYs and OsNACs, inside the nucleus, therefore triggering the expression of functional proteins in response to bacterial pathogens (Figure 2). The activation of upstream signaling cascades leads to the increase of proteins related to the biosynthesis of phenolic phytoalexin, phenylalanine, phenylpropanoids, and flavonoids. Moreover, proteins associated with antioxidation, photosynthesis, pathogenesis-related proteins (PRs), and cell wall modifications were highly accumulated in both intracellular and extracellular spaces (Figure 2). Furthermore, proteins involved in SA biosynthesis were upregulated, whereas ABA signaling was suppressed, suggesting that SA and ABA positively and negatively contribute to bacterial resistance, respectively. However, missing gaps still exist in our systemic understanding of the interactions between rice and bacterial pathogens. Investigations of protein accumulation and PTM dynamics with high-throughput proteomics in combination with multiomics approaches remain goals for the near future.

## 3. Interactions between Rice and Growth-Promoting Bacteria

In addition to pathogens, plants also encounter and are often associated with a variety of non-pathogenic bacteria in the rhizosphere and endosphere regions. These plant-associated bacteria, which may not harm plants and may even promote plant growth under biotic and abiotic stress conditions, are known as plant-growth-promoting rhizobacteria (PGPR) [82,83,84]. Therefore, plant protein changes during the interaction with those microbes may be different from *Xanthomonas.* Thus, several studies have been performed to understand the interaction between rice and plant-growth-promoting bacteria. For instance, *Sinorhizobium meliloti*, a symbiotic rhizobium bacterium in rice, was utilized to investigate its symbiotic relationship with its host. The *S. meliloti* 1021 colonized both the rhizosphere and endosphere in root and leaf tissues and showed significant growth-promoting effects on rice. Additionally, 2-DGE-based proteomics was used to identify the protein changes in the roots and leaves independently [50]. The host defense response mainly took place in root tissue, whereas proteins related to photosynthesis and auxin signaling were upregulated in the shoots. This provides clues for the understanding of how *S. meliloti* 1021 promotes rice growth.

*Stenotrophomonas* maltophilla and *Bacillus* spp. are Gram-negative and Gram-positive bacteria, respectively. A combination of *S. maltophilla* and *Bacillus* spp., which showed a growth-promoting effect, was applied to rice seedlings [51]. Protein changes were determined at 45 days after treatment by the proteomics approach. Among 153 differentially expressed proteins, only 12 could be identified, and the majority of these included uncharacterized proteins. Malate dehydrogenase, HSFB2B, and triosephosphate isomerase were upregulated, suggesting the involvement of glycolysis in plant-growth promotion. Interestingly, the heat shock factor HSFB2B negatively regulated drought and salt tolerance in rice [85], suggesting that HSFB2B may be involved in plant growth and abiotic stress crosstalk in plants.

*Pseudomonas fluorescens* KH-1 also exhibited a growth-promoting effect on rice. Kandasamy and co-workers utilized a proteomics approach to investigate the molecular basis of *P. fluorescens*-induced growth promotion in rice [52]. A total of 23 differentially expressed spots were detected on the 2-DGE map, five of which were identified by LC-MS/MS. The identified proteins included p23 co-chaperone, thioredoxin h, ribulose-bisphosphate carboxylase (Rubisco) large chain, nucleoside diphosphate kinase, proteasome subunit alpha, and putative glutathione S-transferase. These findings suggest that *P. fluoresence* is involved in the growth-promoting process by modulating the host proteins related to energy metabolism, defense, and metabolism to induce systemic resistance against both biotic as well as abiotic stresses response.

Since limited proteins were identified and no further genetic evidence is available at present, the understanding of how PGPRs promote rice growth and defense, especially from the side view of proteomics, is still elusive.

## 4. Interactions between Rice and Fungal Pathogens

*Magnaporthe oryzae* (anamorph *Pyricularia oryzae*) is a hemibiotrophic pathogen that causes rice blast disease. *M. oryzae* invades almost all the tissues of rice, including the leaf, stem, panicle, and root [86,87]. Typically, *M. oryzae* undergoes a biotrophic stage during the initial stages of infection and is involved in the nutrient uptake from the host through invasive hyphae. Once the hyphae spread into the neighbor cells, the initial infected cells undergo cell death. Since rice blast disease is the most devastating fungal disease of rice, several proteomics studies have been carried out to investigate the interaction between rice and *M. oryzae* [53,54,57,88]. The results from these proteomics studies have commonly suggested critical roles of a few proteins, including PBZ1, PR10, RLK/DUF26, SalT, OsIRL, and Cu/Zn-SOD, in rice defense against *M. oryzae* infection.

PBZ1, a member of the PR10 family, exhibits RNase activity and is required for cell death progress in plants [89]. Moreover, both PBZ1 and OsPR10 were reported as defense biomarkers in rice [90]. The overexpression of OsPR10a, also known as PBZ1, in rice significantly enhanced resistance against *Xoo* and *X. campestris* pv. *Campestris* (Xcc) [91]. Furthermore, the overexpression of OsPR10a enhanced rice resistance against *M. oryzae* [91]. Interestingly, the OsPR10a-overexpressing plants showed a significantly increased primary root length under phosphate-deficient conditions, suggesting multiple roles of OsPR10a in defense and nutrient utilization [91]. However, the genetic evidence provided to illustrate the role of this protein in pathogen resistance is still limited.

OsIRL is an isoflavone-reductase-like gene that is required for the biosynthesis of chiral pterocarpan phytoalexins. The expression of OsIRL was significantly induced by rice blast elicitor and JA treatments but was suppressed by SA and ABA [54]. The overexpression of OsIRL enhanced oxidative stress tolerance in rice [92]. In soybeans, the overexpression of isoflavone reductase enhanced the resistance against *Phytophthora sojae* [93]. However, since no evidence have been shown that chiral pterocarpan phytoalexins are synthesized in rice, the biochemical activity of OsIRL has not been analyzed. Therefore, how OsIRL contributes to plant immunity is still unclear.

Receptor-like/Pelle kinases (RLKs) are conserved signaling components that contribute to plant development and responses to biotic/abiotic stress conditions. The domain of unknown function 26 (DUF26) containing RLKs are the plant-specific proteins containing a cysteine-rich motif. They are also known as cysteine-rich receptor-like kinases (CRKs) or *O. sativa* root meander curling (OsRMC) [94,95]. A previous 2-DGE-based proteomics study showed that OsRMC was detected in the rice apoplast under salt-stress conditions, suggesting the possible role of DUF26 in apoplastic resistance [96]. The knock-down of OsRMC enhanced salt-stress tolerance of the transgenic lines, suggesting a negative role of OsRMC in abiotic stress tolerance in rice. Moreover, OsRMC knockdown lines showed an enhanced JA-sensitivity in the roots [97]. The expression of OsRMC was regulated by two transcription factors, Ethylene-Responsive Element Binding Protein 1 (OsEREBP1) and OsEREBP2, under salt-stress conditions [98]. These findings highlight the possible role of DUF26 in ethylene- and JA-mediated signaling in rice. A recent study showed that OsRMC is required for resistance against rice blast fungus through a direct interaction with the fungal-secreted carbohydrate-binding module 1 (CBP1) [99].

Recently, an iTRAQ-based proteomics analysis was performed during the compatible and incompatible interactions between rice and *M. oryzae* [58]. Among 4154 identified proteins, 193 and 672 differential proteins were identified in the resistant and susceptible rice varieties, respectively. Proteins involved in incompatible interactions were mainly related to plant–pathogen interactions, hormone signaling, fatty acid metabolism, and peroxisome. Moreover, a group of proteins related to defense was significantly upregulated during the incompatible interaction with *M. oryzae*. These differentially accumulated proteins included receptor kinases (RLCK, CDPK, and MAPK), defense-related proteins (glucanases, chitinases, OsPR9, PBZ1, OsPR10, and OsPR10a), and ROS-related proteins (ascorbate peroxidases and peroxidases). Interestingly, an OsFLS2-like protein was increased in the incompatible interaction and reduced during the compatible interaction. OsCEBiP, the first chitin-binding protein identified in rice, triggers immune responses through the OsCEBiP/OsCERK1-OsRacGEF1-OsRac1 signaling cascade [35,100]. During rice blast infection, the OsCEBiP was not changed in an incompatible interaction but was inhibited in the compatible interaction [58]. These findings suggest the importance of membrane receptors, which mediate the activation of PTI in rice immunity against fungal pathogens.

Kang and colleagues investigated the protein profiles in the culture medium of rice-suspension-cultured cells in response to *M. oryzae* spores or an elicitor treatment [55]. Proteins including chitinase, DUF26, germin A, expansin, and amylase were highly accumulated in the culture medium [55]. Subsequently, 2-DGE-MS/MS and MudPIT approaches were employed to investigate the in planta apoplastic protein changes upon rice blast infection. This led to the identification of 291 unique proteins related to stress response, ROSs, and energy metabolism [56]. Several isoforms of chitinases, DUF26, peroxidases, PR1s, and PR5s were highly accumulated, while glycosyl hydrolases (GHs), protein inhibitors, and peptidases/proteases were only detected during in planta growth conditions. GHs are involved in the modifications of cell wall, which is closely related to disease resistance responses and rice development [101,102].

*M. oryzae* snodprot1 homolog protein (MSP1), a secreted protein from rice blast fungus, was reported to induce PTI in rice [103]. A TMT-based proteomics revealed the importance of plasma-membrane-localized RLKs and the co-receptor BAK1 in the perception of this fungal signal [67]. MAPK13, which was upregulated upon treatment with benzothiadiazole (a synthetic analog of SA) and *M. oryzae* [67], was significantly increased in the extracellular MSP1-expressing plants, suggesting that MAPK13 may be involved in MSP1-induced immunity. Phospholipase, the lipolytic enzyme that hydrolyzes phospholipid substrates, is required for signal transduction in eukaryotes. An increase of OsPLC was observed in rice overexpressing MSP1 [104]. The previous result showed that OsPLC1 was significantly activated in a systemically acquired resistance in rice [105]. However, the molecular mechanism of how MAPK13 and OsPLC contribute to rice immunity is still elusive.

In order to investigate the changes in the phosphoproteome of resistant and susceptible rice cultivars upon *M. oryzae* infection, an Al(OH)3-MOAC-based approach was utilized for the enrichment of phosphoproteins [106]. A total of 53 significantly regulated phospho-spots were identified and functionally annotated. This demonstrated that the phosphorylation levels of proteins related to photosynthesis and redox states were dramatically repressed in both compatible and incompatible interactions. Moreover, the phosphorylation of proteins involved in signaling and microtubule-based processes were differentially accumulated. It was reported that the microtubule was aggregated in the region of fungal infection [107], and treatment with actin cytoskeleton depolymerizing chemicals resulted in the inhibition of cytoplasmic aggregation, papilla formation, hypersensitive response, and defense gene activation in plants, converting the resistant plant into a plant susceptible to the invasion/infection of pathogens [108,109]. These results suggest the possible role of microtubule-based processes in pathogen resistance. However, it was reported that the microtubule-related processes functioned differently during the infection of different microbes [107]. For instance, formations of specific microtubule arrays were detected during the development of infection threads and hyphae with rhizobia and mycorrhizal fungus, whereas bacterial elicitors and effectors affected the plant microtubular cytoskeleton [107]. The phosphorylation of a WRKY transcription factor, OsWRKY11, was detected specifically in the incompatible host cells, suggesting a possible role of OsWRKY11 in rice blast resistance. Further study showed that OsWRKY11 regulates the expression of defense-related gene Chitinase2 and the drought resistance gene RAB21 by directly binding to their promoters, contributing to the resistance against *Xoo* and drought, respectively [110]. However, both the overexpression and knockdown of OsWRKY11 significantly reduced rice growth [110]. Therefore, the molecular mechanism of OsWRKY11 in the defense–growth trade-off still needs to be addressed.

*Rhizoctonia solani* and *Fusarium fujikuroi* are both necrotrophic fungal pathogens of rice. *R. solani* infects the root and leaf sheath, while *F. fujikuroi* infects the panicle at the flowering stage to cause sheath blight and bakanae disease, respectively. To understand the protein changes during *R. solani* infection, 2-DGE-MS/MS-based proteomics was employed [59]. A total of 17 identified spots related to defense, protein degradation, and photosynthesis were induced upon *R. solani* infection. Interestingly, GAPDH and 3-β-hydroxysteroid dehydrogenase (HSD) were highly accumulated in the incompatible interaction, suggesting the possible role of the glycolytic pathway and steroid metabolite in necrotrophic fungal resistance. Similarly, to dissect the molecular mechanism of rice resistance against *F. fujikuroi*, Ji et al. performed a TMT-based quantitative proteomics analysis in rice. Rice young seedlings of the *Japonica* genotype (Nipponbare) and *indica* genotype (9311), which are susceptible and resistant to *F. fujikuroi*, were used to determine the differentially expressed proteins [60]. Plasma Membrane Intrinsic protein 2-2 (PIP2-2) and vacuolar-sorting receptor 3 (VSR3) were highly accumulated in the resistance cultivar, suggesting their putative roles in resistance. PIPs are aquaporin proteins that control the water balance in plants. In rice, the PIP family contains 11 members which perform crucial functions in biotic and abiotic stress responses [111,112]. In Arabidopsis, PIP1;4 was involved in the transport of H_2_O_2_ from the apoplast to the cytoplasm and positively contributed to plant immunity [113]. Interestingly, the ectopic expression of apple (*Malus domestica*) MdPIP1;3 significantly increased the fruit size in tomatoes, suggesting a possible role of PIPs in both plant immunity and development [114].

The necrotrophic fungal pathogen *Cochliobolus miyabeanus* causes brown spot disease and infects the leaf tissues in rice [61]. To investigate the changes in protein profiles and to identify the low-abundance signaling-related proteins, Kim and co-workers utilized a PEG fractionation method for protein extraction. This method enhances the detection of low-abundant proteins [115]. Based on the proteomics results, proteins related to metabolic and oxidation/reduction were highly accumulated during *C. miyabeanus* infection. Moreover, the abundance of proteins related to cell redox, the TCA cycle, amino acids, and ethylene-related proteins was significantly increased, whereas the Calvin cycle- and glycolysis-related proteins were suppressed. Furthermore, PR proteins (β-1,3-glucanase, PR10, SalT, and TLP) were significantly upregulated. This was further validated by a Western blot analysis. In addition, the abundance of ET biosynthesis protein homocysteine S-methyltransferase was significantly increased upon *C. miyabeanus* infection. This was consistent with the results of a subsequent study in which it was shown that infection with *C. miyabeanus* significantly increases ethylene production in rice [116]. The suppression of OsEIN2a significantly reduced the infection of *C. miyabeanus*, suggesting that *C. miyabeanus* employs the host ethylene pathway to promote its infection [116]. Subsequently, a shotgun proteomics approach (SDS-PAGE coupled with nESI-LC-MS/MS) was employed to identify the secreted proteins from both rice and *C. miyabeanus*; thisled to the identification of 470 and 31 proteins, respectively. A group of secreted proteins related to protein degradation (aspartic protease, subtilisin, and serine carboxypeptidase), ROS metabolism (peroxidases and monodehydroascorbate reductase), and cell wall modifications (β-glucosidase, chitinases, alpha-amylase, cellulase, and alpha-N-arabinofuranosidase) was highly accumulated in the apoplast during *C. miyabeanus* infection. In Arabidopsis, the secreted aspartic protease suppressed bacterial growth through direct targeting of bacterial growth-related protein [117].

Protein ubiquitination, which leads to proteome-mediated degradation, is a key regulatory mechanism for plant immunity. A significant increase in the protein ubiquitination levels was obseved in rice cells by Western-blotting analysis after challenging with PAMP elicitors, chitin and flg22 [17]. As expected, the ubiquitination level of the ubiquitination system was increased. Interestingly, protein transportation, ligand recognition, membrane trafficking, and redox-reaction-related proteins were highly ubiquitinated. It was reported that the recycling of proteins involved in ligand recognition and membrane trafficking was required for the full activation of plant immunity [118,119]. Therefore, the ubiquitination and degradation of protein-transportation-related proteins may also be required for the attenuation of innate plant immunity. Moreover, the ubiquitination of phenylpropanoid-biosynthesis-related proteins was highly elevated, suggesting that the phenylpropanoid metabolites may suppress pathogen infection directly or indirectly. Furthermore, among proteins with different ubiquitination levels, less than 30% were shared between two types of elicitors, which indicates that the signal transduction between bacterial and fungal infection is largely diverse in rice. Interestingly, the ubiquitination levels of proteins related to translation and signal transduction were significantly reduced, indicating the importance of those processes in broad resistance.

To gain a broader insight into rice immunity at the multiomics level, a systemic study was further performed by employing the transcriptome, proteome, ubiquitin, acetylome, and metabolome [18]. Interestingly, although the phenylpropanoid biosynthesis and flavonoid biosynthesis pathways were suppressed at the RNA and protein levels, the metabolite accumulation was significantly increased [18]. This may be due to the high level of protein ubiquitination and acetylation of enzymes involved in the secondary metabolite biosynthesis process. We also noticed a correlation among the RNA, protein, PTM, and metabolite levels from this multiomics approach. This may be because the accumulation of metabolites is much more delayed compared with RNA and protein level changes. In this study, Tang et al. collected all samples at the same time post-elicitor treatment. Therefore, a time-dependent multi-omics analysis could provide an intact profile of rice immune responses systemically. Interestingly, the acetylation level of OsWRKY30 was significantly increased upon chitin and flg22 treatment [18]. In animals, the acetylation of proteins enhances or suppresses the DNA-binding activity of nonhistone proteins and transcription factors [120]. It was reported that the OsWRKY30 was regulated by OsMKK3-OsMPK7 and OsMPK6 in rice resistance against *Xoo* and *Xoc*, respectively [121,122]. Taken together, the acetylation of OsWRKY30 may benefit from its DNA binding activity for the regulation of downstream gene expression. However, how protein phosphorylation and acetylation coordinate to regulate the function of transcription factors and their PTM dynamics upon pathogen infection are still unaddressed.

Taken together, in response to fungal pathogens, the phosphorylation of OsCERK1, OsMAPKs, and downstream transcription factors such as OsWRKYs and OsEREBPs are required for perception and signal transduction (Figure 2). Ethylene signaling and water control through water channel proteins may also be required for resistance against fungal pathogens. The accumulation of PR proteins and ROS-detoxifying proteins is essential for fungal pathogen resistance. Furthermore, cell-wall-modification enzymes and protein-degradation-related proteins that are highly accumulated in the apoplastic region may also be required for fungal resistance in rice. However, the exact role and molecular mechanisms of their action still need to be investigated.

## 5. Interactions between Rice and Virus Pathogens

Proteomics studies have also been performed to study the interaction between rice and different viruses including rice yellow mottle virus (RYMV), rice stripe virus (RSV), rice black-streaked dwarf virus (RBSDV), and southern rice black-streaked dwarf virus (SRBSDV) (Table 1).

The RYMV, a member of the genus Sobemovirus, was isolated on the African continent and protein changes in susceptible and resistant rice cultivars were analyzed [62]. Redox-state-related proteins such as superoxide dismutase (SOD), heat-shock proteins (HSPs), abiotic-stress-related proteins such as RAB25, and salt-stress-induced protein (SALT) were upregulated, whereas the ethylene-inducible protein was suppressed, suggesting that ethylene signaling may play a crucial role in RYMV resistance.

RSV, a member of the genus *Tenuivirus*, is widespread in East Asian countries. It is transmitted by the small brown planthopper in the field. An iTRAQ-based proteomics analysis was performed which led to the identification of 358 proteins differentially accumulated upon RSV infection [63]. Further analysis indicated that plant defense (PR10, PR1, pathogenesis-related protein, and bet v I allergen family protein) and ROS-related proteins (SOD and peroxidases) were significantly increased. In contrast, chlorophyll biosynthesis and photosynthesis-related proteins were significantly downregulated, which is consistent with the leaf chlorosis phenotype caused by RSV.

RBSDV and SRBSDV both belong to the genus Fijivirus in the family Reoviridae. RBSDV infects rice and maize and leads to rice black-streaked dwarf disease and maize rough dwarf disease, respectively. An infection of RBSDV induced the production of H_2_O_2_ in the susceptible rice cultivar [123]. Among 1800 protein spots, 69 DEPs were identified. The upregulated proteins were related to defense and stress response, whereas the photosynthesis-related proteins were downregulated. Interestingly, the DUF26 protein, which is involved in rice blast resistance, was significantly increased. However, the biochemical function of DUF26 is still unclear. Likewise, protein changes in the resistant and susceptible rice cultivars were analyzed with and without SRBSDV infection [65]. The most highly regulated proteins upon SRBSDV infection in resistant and susceptible rice cultivars were related to systemic acquired resistance (SAR). In addition, PR proteins such as PR1, PR10, and PR3 were highly accumulated in the resistant cultivar after viral infection.

Chitosan oligosaccharide, considered as a potent elicitor, induces a host immune response in many plants. Cytosinpeptidemycin is a microbial pesticide that displays broad-spectrum antiviral activity against various plant viruses. However, the molecular mechanisms underlying their antiviral activity are poorly understood. A proteomics approach was utilized to understand the protein level changes in rice with the treatment of chitosan oligosaccharide and cytosinpeptidemycin, respectively [64,66]. Proteins related to defense and ROS detoxification, such as PODs, SODs, and CATs, were significantly upregulated, suggesting their role in plant defense against viruses.

These findings illustrate the importance of ROS-detoxifying proteins in viral disease resistance in rice. The consistent overexpression of miR528 negatively regulates viral resistance in rice by cleaving the mRNA of the ROS-detoxifying protein ascorbate peroxidase [124]. However, it was known that high level of ROSs significantly suppress plant growth [125]. Therefore, maintaining ROS levels with ROS-detoxifying proteins may benefit the breeding of rice with a high resistance and better performance.

## 6. Summary and Perspectives

In this review, we summarized the progress of rice protein changes upon interaction with pathogenic and commensal bacteria, fungi, and viruses. An array of signaling networks and key proteins involved in rice immunity have been identified through large-scale proteomics approaches, including proteins related to signal transduction, ROS homeostasis, and plant defense (Table 1). A group of identified rice immune components is shared during the interaction between different types of microbes (Figure 3), which provides an opportunity to develop crops that are resistant to a broad range of pathogens.

The MAPK signaling cascades and calcium signaling regulated CDPKs play a key role in the regulation of downstream transcription factors, especially WRKYs. They are required for the activation of a defense response in both bacterial and fungal pathogens (Figure 2). The redox-state-related proteins and PRs are the most highly accumulated proteins upon infection with bacteria, fungi, and viruses, suggesting that a balance of redox states inside the cell is essential for immune response in the host. Moreover, proteins related to GHs and cell wall modifications are highly induced and accumulated in the apoplastic region. Compared with bacterial infection, protein-degradation-related proteins were highly accumulated in the apoplastic region upon fungal infection, suggesting that the cleaving of fungal-derived proteins may be a stratagem in the plant for fungal resistance. Secondary metabolite biosynthesis pathways were also involved in the response against pathogen infection. Phenylalanine and phenylpropanoids biosynthesis were highly induced by both bacterial and fungal pathogens, indicating that these metabolites may have a broader role in pathogen resistance. Bacteria lead to an increase in flavonoids and phytoalexin biosynthesis, whereas *M. oryzae* infection leads to the increase of protein abundance of ascorbate and aldarate and amino sugar and nucleotide sugar biosynthesis-related proteins (Figure 2). The phosphorylation of PP2Cs negatively contributes to ABA signaling and, in turn, enhances plant immunity against bacterial infection process. An increase of prohibitin and ICS1, which are localized in plant chloroplasts, were detected. They contribute to bacterial resistance by enhancing SA biosynthesis. OsArg1, a mitochondria-localized protein, was also involved in rice resistance against the bacterium *Xoo*; however, the molecular mechanism still needs to be addressed. In response to the fungal pathogen, the accumulation of ET biosynthesis-related protein homocysteine S-methyltransferase (HSM) increased upon fungal infection. A plasma-membrane-localized water transporter PIP2-2 accumulated upon bacterial infection, illustrating that water balance is also required for host resistance against fungal pathogens. Moreover, microtubule-structure-related proteins were alternated during *M. oryzae* infection, suggesting the involvement of cell structural proteins in fungal pathogen resistance in rice.

Compared to the well-established transcriptome studies, limited proteome studies have been carried out to date. In particular, gene knockout mutants have not been utilized for the proteome investigations. Moreover, plant responses under different types of immunity are diverse in dynamics. PTI triggers a rapid and transient immune response, whereas ETI leads to a more robust response but is initiated relatively slowly. Therefore, the time-dependent experiment for investigating protein abundance dynamics has still not been well-investigated, limiting our understanding of plant immunity. The PTM of proteins greatly expands proteome diversity. The PTM changes and crosstalk between different protein PTMs during rice–microbe interactions have been poorly investigated. It has been shown that PTMs frequently take place on the metabolic enzymes, which affect the enzyme kinetics, stability, and activity. Therefore, a combination of protein quantification, modification, and metabolomic approaches would strongly benefit our systemic understanding of the plant–microbe interaction.

Recently, antibody-array-based proteomics tools were developed and successfully applied in mammalian and plant research [126]. The application of this antibody-array-based proteomic approach may provide novel information on plants during pathogen invasion [127]. Moreover, the generation of genome-wide antibody resources provides the possibility of investigating protein–protein interaction networks using a combination of immunoprecipitation and MS identification approaches. Another limitation of omics studies is that only bulk samples could be used to understand overall changes in a mixed cell type. The development of a method for low-input identification in combination with reporter lines harboring specific reporters provides the potential for the establishment of single-cell proteomics. These highlighted methods of proteomics may provide opportunities for future research on plant–microbe interactions.

## Figures and Tables

**Figure 1 plants-12-01079-f001:**
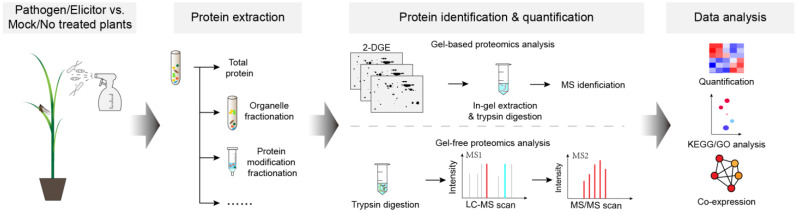
Workflow of proteomics investigation of plant–microbe interactions. Rice plants are inoculated with bacterial, fungal, and viral pathogens through a spray, syringe infiltration, friction, and/or insect-mediated inoculation. Infected tissues are used for protein extraction. In additional to total protein extraction, organelle fractionation, enrichment of post-transcriptional modifications, and other extraction methods are being used to improve the resolution and detection of low-abundance proteins or proteins with different modifications. Extracted proteins are then subjected to protein identification and quantification with gel-based and gel-free proteomics analyses. In the gel-based proteomics approach, protein samples are separated by two-dimensional gel electrophoresis (2-DGE), and protein spots are quantified with in silico methods. The differentially expressed protein spots are identified after in-gel extraction and digestion. For the gel-free proteomics approach, proteins are directly digested and identified by MS/MS approach. Bioinformatic approaches such as heatmaps, KEGG/GO analysis, and co-expression network establishment are further performed for the graphical representation of the obtained data.

**Figure 2 plants-12-01079-f002:**
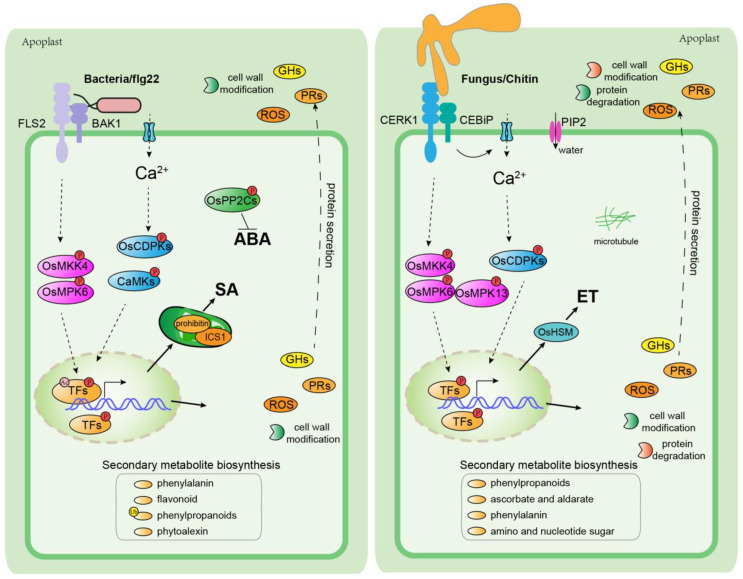
Proteomics-based schematic diagram of rice–microbe interactions. Sensing of bacterial and fungal pathogens by membrane-localized pattern recognition receptors leads to the phosphorylation of MAPK cascade and CDPKs in rice, which subsequently activates the downstream transcription factors, especially WRKKYs. The abundance of glycoside hydrolase family proteins (GHs), reactive-oxygen-species-related proteins (ROSs), pathogenesis-related proteins (PRs), cell-wall-modification-related proteins, and protein-degradation-related proteins are significantly increased and highly accumulated in the apoplastic region through protein secretion. Secondary metabolite biosynthesis-related proteins are also highly accumulated upon bacterial (left panel) and fungal pathogen (right panel) infection. Accumulation of SA and ET biosynthesis regulating proteins prohibitin, ICS1, and HSM were increased upon bacterial and pathogen infection, respectively. Phosphorylation of PP2Cs, a negative regulator of ABA signaling, is increased upon bacterial infection.

**Figure 3 plants-12-01079-f003:**
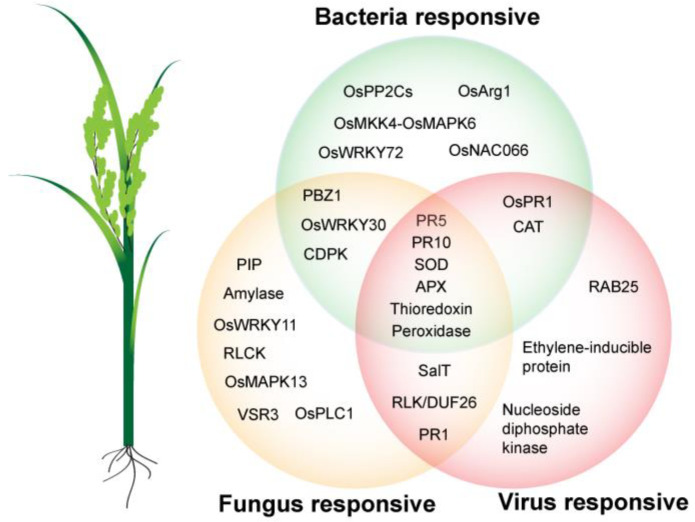
Proteomics investigation of rice–microbe interactions. Interactions between rice and different types of microbes, including pathogenic bacteria, symbiotic bacteria, fungi, and viruses, have been investigated with proteomics approaches in recent decades. Proteome results indicate that the ascorbate peroxidase (APX), superoxide dismutase (SOD), peroxidases, pathogenesis-related (PR) protein PR5, PR10, and thioredoxin are identified in response to bacteria, fungi, and viruses, while probenazole-induced protein (PBZ1), OsWRKY30, and calcium-dependent protein kinase (CDPK) are responsive only to bacterial and fungal pathogens. Salt-induced protein (SalT), receptor-like kinase/domain of unknown function 26 (RLK/DUF26), and PR1 are induced upon fungal and viral infections. OsPR1, heat-shock proteins (HSPs), and catalase [21] are induced by bacteria and viruses. A group of proteins specific to bacteria, fungi, and viruses has also been identified. OsPP2C, protein phosphatase 2C; OsMKK4, mitogen-activated protein kinase kinase 4; OsMPK6, mitogen-activated protein kinase 6; VSR3, vacuolar sorting receptor 3; PIP, plasma membrane intrinsic proteins; RAB25, and ras-related protein 25.

**Table 1 plants-12-01079-t001:** Proteomics investigations of rice–microbe interactions.

Microbes/Elicitors		Method	Rice Cultivar	Sample	Differentially Accumulated Proteins	References
Bacterium	*Xanthomonas oryzae* pv. *oryzae (Xoo)*	2-DGE, MALDI-TOF MS	*Oryza sativa* L. cv. Java 14	Leaf cytoplasm, membrane protein	PBZ1, PR5, SOD, Peroxiredoxin	[40]
		2-DGE, MALDI TOF-TOF MS/MS	*Oryza sativa* L. *japonica* cv Nipponbare harboring Xa21-GFP	Leaf plasma membrane protein	PM-associated H+-ATPase, Protein phosphatase, Hypersensitive-induced response protein, Prohibitin	[41]
		2-DGE, MALDI-TOF-TOF	*Oryza meyeriana* L.	Leaf total protein	Ascorbate peroxidase, putative Glutathione S-transferase, Mitochondrial chaperonin-60	[42]
		2D-DIGE, MALDI-TOF-TOF	*O. sativa* L. *japonica* cv Nipponbare	Secreted protein from suspension-cultured cells	Cu/Zn-SOD, Cellulase, CHIT16	[43]
		2D-DIGE, MALDI-TOF-TOF	*Oryza meyeriana* L.	Secreted protein from suspension-cultured cells	Ser/Thr protein phosphatase family protein, Phospholipase C, GDSL-like lipase/acyl hydrolase, OsPDIL1-1, Glucan endo-1,3-beta-glucosidase, Peroxidases, Cu/Zn-SOD, Expansin	[44]
		SDS-PAGE, MudPIT	*Oryza sativa* L. *japonica* cv Dongjin	Secreted protein from suspension-cultured cells and leaves	Peroxidase, Peroxiredoxin, Cu/Zn-SOD, Ferrodexin, Glutathione S-transferase, Thioredoxin, Ascorbate peroxidase, Chitinase, Thaumatin-like proteins, Pathogenesis-related bet VI family protein	[45]
		TiO2-MOAC, nLC-MS/MS	*Oryza sativa* L. cv. IRBB5 IRBB13	Leaf total protein	PP2Cs, Brassinosteroid insensitive 1-associated receptor kinase 1, OsWRKY72	[46]
		TMT, LC-MS/MS	Rice introgression line	Leaf total protein	CDPK13, OsMKK4, OsMPK6, OsPR1b	[47]
		nLC-MS/MS	*O. sativa* L. *japonica* cv Dongjin, Hwayeong	Low-abundance protein from leaves	CDPKs, PTI-like tyrosin-protein kinase, serine/threonine-protein kinase, OsArg1	[48]
	*Xanthomonas campestris* pv. *oryzicola (Xoc)*	2-DGE, MALDI-TOF-MS	*Oryza sativa* L. *indica* cv. 9311	Leaf total protein	OsMPK6, Allene oxide synthase 3, receptor-like kinase, L-ascorbate peroxidase 3, PR1-like protein, PR10	[49]
	*Sinorhizobium meliloti*	2-DGE, MALDI-TOF-MS	*O. sativa* L. *japonica* cv Nipponbare	Root, leaf sheath, leaf total protein	Subtilisin-like proteinase, Exoglucanase, Enolase, Catalase, Auxin-induced protein	[50]
	*Stenotrophomonas maltophilla and bacillus*	2-DGE, nLC-MS/MS	*Oryza sativa* L. *indica* cv. MR219-9	Leaf sheath total protein	Malate dehydrogenase, HSFB2B, Triosephosphateisomerase	[51]
	*Pseudomonas fluorescens*	2-DGE, nLC-MS/MS	*Oryza sativa* L. *indica* cv. CO43	Leaf sheath total protein	Thioredoxin, Nucleotide Diphosphate kinase, putative glutathione S-transferase	[52]
Fungus	*Magnaporthe oryzae (Mo)*	2-DGE, N-terminal, and internal amino acid sequence analysis	*Oryza sativa* L. cv Hitomebore	Leaf sheath total protein	Oxygen-evolving enhancer protein 2, Fe-SOD, Cu/Zn-SOD, Thaumatin-like protein	[53]
		2-DGE, N-terminal, and internal amino acid sequence analysis	*Oryza sativa* L. *japonica* cv. Jinheung	Leaf total protein	PBZ1, SalT, β-Glucosidase, OsIRL, PR10	[54]
		2-DGE, MALDI-TOF-MS	*Oryza sativa* L. *japonica* cv. Jinheung	Suspension-cultured cell secreted protein	Chitinases, DUF26s, α-Amylases, Germin A	[55]
		2-DGE, MALDI-TOF-TOF, nESI-LC-MS/MS	*Oryza sativa* L. *japonica* cv. Jinheung	Leaf-secreted protein	Xylanase inhibitors, GH family proteins, DUF26s, PR5s, chitinases, PR1s, Proteases, Peroxidases	[56]
		2-DGE, MALDI-TOF-MS	*Oryza sativa* L. *japonica* cv. Jinheung	Leaf total protein	PBZ, PR10, β-1,3-glucanase1, β-1,3-glucanase2, TLP, RLK, POX22.3	[57]
		iTRAQ, LC-ESI-MS/MS	*Oryza sativa* L. cv Gangyuan8, Lijiangxintuanheigu	Leaf total protein	Defense proteins (PR1s, PR2s, PR3s, PR8s, PR10s, PR14s, PR15s), redox-oxygen-species-related proteins (Peroxidases, apxs), receptor kinases (DUF26s, RLCKs, LRRs, CDPK, MAPK)	[58]
	*Rhizoctonia solani*	2-DGE, ESI-Q-TOF MS	*Oryza sativa* L. cv Labelle, LSBR-5	Leaf sheath total protein	β-1,3-glucanase, Stomatal ascorbate Peroxidase, Chitinase, 14-3-3 like protein	[59]
	*Fusarium fujikuroi*	TMT labeling, LC-MS/MS	*Oryza sativa* L. cv Nipponbare, 9311	Seedling total protein	PIP2, Peroxidase, PR1, SBT3.8, Monodehydroascorbate reductase, Salicylic acid-binding protein2	[60]
	*Cochliobolus miyabeanus*	2-DGE, MALDI-TOF-TOF, nESI-LC-MS/MS	*Oryza sativa* L. *japonica* cv. Jinheung	Leaf total and secreted proteins	β-1,3-glucanase, Chitinase, Cu/Zn-SOD, Glutathione reductase, Thioredoxin, Protein disulfide isomerase	[61]
Virus	*Rice yellow mottle virus (RYMV)*	2-DGE, MALDI-TOF-MS, nLC-MS/MS	*Oryza sativa* L. cv IR64, Azucena	Suspension-cultured cell total protein	Cu/Zn-SOD, α-amylases, HSP70s, Ethylene-inducible protein, PR10a,	[62]
	*Rice stripe virus (RSV)*	iTRAQ, LC-MS/MS	*Oryza sativa* L. cv Aichiasahi	Leaf total protein	PR1, PR10, Ascorbate peroxidase 1, Thioredoxin, Cu/Zn-SOD, Mn-SOD, Peroxidases	[63]
	*Rice black-streaked dwarf virus (RBSDV)*	SDS-PAGE, nLC-MS/MS	*O. sativa* L. *japonica* cv Nipponbare	Leaf total protein	PP2A, Glycolate oxidase1, Glycolate oxidase 5, Peroxidases, Catalase, Nucleoside diphosphate kinase	[64]
		SDS-PAGE, nLC-MS/MS	*Oryza sativa* L. cv Z1, L2186, FYXZ	Leaf total protein	Dolichyl-diphosphooligosaccharide-protein glycosyltransferase, Hypoxia upregulated protein, Membrane-attack complex	[65]
	*Southern rice black-streaked dwarf virus (SRBSDV)*	SDS-PAGE, nLC-MS/MS	*O. sativa* L. *japonica* cv Nipponbare	Leaf total protein	PR5, PR10, PODs, SODs, CAT	[66]
Elicitors	Chitin, flg22	Affinity enrichment of ubiquitinated peptide, LC-MS/MS	*O. sativa* L. *japonica* cv Nipponbare	Seedlings	ubiquitination system, protein transportation, ligand recognition, membrane trafficking, redox reactions, phenylpropanoid metabolic	[17]
	Chitin, flg22	Total, affinity enrichment of ubiquitinated and acetylated peptides, LC-MS/MS	*O. sativa* L. *japonica* cv Nipponbare	Seedlings	Enzymes involved in secondary metabolite biosynthesis, WRKY30	[18]
	MSP1 (PTI-inducing protein secreted from *M. oryzae*)	TMT labeling, LC-MS/MS	*O. sativa* L. *japonica* cv Dongjin	Leaf total protein	RLKs, BAK1, MAPK, CRT	[67]

## Data Availability

Not applicable.

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
