# Peer review of "A Proteomics Insight into Advancements in the Rice–Microbe Interaction"

_plants, 2023, doi:10.3390/plants12051079_

Round 1

Reviewer 1 Report

Rice is one of the most crops. This review helps us to understand the complexity of the rice-pathogen interaction. Some questions as follows:

(1) Proteomics approach is a part of this review. Therefore, suggesting that authors revise this title.

(2) In figure 2, some proteins should be added “Os”. Additionally, some transcription factor families are still important except for WRKY, suggesting that authors use TFs to represent all TFs, or list 2-3 TF families. Please make corresponding modifications in the MS.

(3) In figure 3, suggesting that authors cite some key genes from recent published papers. Please make corresponding modifications in the MS.

(4) There are some errors in singular and plural, grammar. Please check and revise them.

Author Response

Rice is one of the most crops. This review helps us to understand the complexity of the rice-pathogen interaction. Some questions as follows:

  • Proteomics approach is a part of this review. Therefore, suggesting that authors revise this title.

Authors: Thanks for your suggestion. We alternated the title to be “A Proteomics Insight into the Advancements in the Rice-Microbe Interaction”

  • In figure 2, some proteins should be added “Os”. Additionally, some transcription factor families are still important except for WRKY, suggesting that authors use TFs to represent all TFs, or list 2-3 TF families. Please make corresponding modifications in the MS.

Authors: We changed the TFs instead of WRKY in the figure. And the manuscript was also modified as we mentioned (Line 312-315 and Line 562-563).

  • In figure 3, suggesting that authors cite some key genes from recent published papers. Please make corresponding modifications in the MS.

Authors: We removed some proteins without significant improvement of understanding of plant-microbe interactions, and also modified in the MS.

(4) There are some errors in singular and plural, grammar. Please check and revise them.

Authors: We have now revised the text to avoid the grammatical errors.

(5) This review outlines recent studies investigating interactions between rice plants and microorganisms using proteomic approaches. A similar review (Ref. 25) was published in 2021, but it is written from a somewhat different perspective and should be of some value as a new review for researchers in plant-microbe interactions.

Authors: Thank you for your positive-feedback on our article.

Reviewer 2 Report

This review outlines recent studies investigating interactions between rice plants and microorganisms using proteomic approaches. A similar review (Ref. 25) was published in 2021, but it is written from a somewhat different perspective and should be of some value as a new review for researchers in plant-microbe interactions.

Please consider the following suggestions to improve the manuscript:

Lines 25-27

This sentence is too definitive about the dangers of pesticides, even though the pesticides used in recent years should have passed various safety tests. Of course, pesticides have various risks, but it would be better to use milder descriptions.

Line 48 and others

The International Mycological Association recommends using Pyricularia oryzae rather than Magnaporthe oryzae. At least, Pyricularia oryzae should be listed as a synonym.

Zhang, N.; Luo, J.; Rossman, A.Y.; Aoki, T.; Chuma, I.; Crous, P.W.; Dean, R.; de Vries, R.P.; Donofrio, N.; Hyde, K.D.; Lebrun, M.-H.; Talbot, N.J; Tharreau, D.; Tosa, Y.; Valent, B.; Wang, Z.; Xu, J.-R. Generic names in Magnaporthales. IMA Fungus 2016, 7, 155-159, doi:10.5598/ imafungus.2016.07.01.09.

Line 185.

Please do not use the abbreviation "TMT" without a definition. There may be other abbreviations that should be defined before use.

 Table 1

The heading "Pathogens" is incorrect because it includes growth-promoting bacteria.

Lines 377-378

This sentence is misleading. Rice is unknown to biosynthesize isoflavonoids or pterocarpan. I assume that the function of OsIRL is unknown and is a homolog of isoflavone reductase, which is involved in the biosynthesis of pterocarpan phytoalexins in legumes.

Line 509.

Is "PAPM" a typo for "PAMP"?

Lines 752-753

The citation of Ref. 33 is wrong. Please check the other references carefully again, as I just found this wrong citation by chance.

Zhang, F.; Huang, L.; Zeng, D.; Cruz, C.V.; Li, Z.; Zhou, Y. Comparative proteomic analysis reveals novel insights into the interaction between rice and Xanthomonas oryzae pv. oryzae. BMC Plant Biol. 2020, 20, 563, 10.1186/s12870-020-02769-7.

Author Response

  • Lines 25-27. This sentence is too definitive about the dangers of pesticides, even though the pesticides used in recent years should have passed various safety tests. Of course, pesticides have various risks, but it would be better to use milder descriptions.

Authors: Thank you for your suggestion. We have now modified the sentence as “Although pesticides increase crop productivity by limiting infection, extensive and continuous use of pesticides can be a threat to the environment because of their bio-magnification and persistent nature”. We hope that the modified sentence justifies the reviewer’s concern.

  • Line 48 and others. The International Mycological Association recommends using Pyricularia oryzae rather than Magnaporthe oryzae. At least, Pyricularia oryzae should be listed as a synonym.

Zhang, N.; Luo, J.; Rossman, A.Y.; Aoki, T.; Chuma, I.; Crous, P.W.; Dean, R.; de Vries, R.P.; Donofrio, N.; Hyde, K.D.; Lebrun, M.-H.; Talbot, N.J; Tharreau, D.; Tosa, Y.; Valent, B.; Wang, Z.; Xu, J.-R. Generic names in Magnaporthales. IMA Fungus 2016, 7, 155-159, doi:10.5598/ imafungus.2016.07.01.09.

Authors: We added this information in the manuscript (line 50-51).

  • Line 185. Please do not use the abbreviation "TMT" without a definition. There may be other abbreviations that should be defined before use.

Authors: Thank you for your suggestion, the definition of TMT was mentioned in line 100 with tandem mass tags (TMT). We also check other observations.

  • Table 1. The heading "Pathogens" is incorrect because it includes growth-promoting bacteria.

Authors: Thank you for pointing out the error. We have now changed “Pathogens” to “Microbes”.

  • Lines 377-378. This sentence is misleading. Rice is unknown to biosynthesize isoflavonoids or pterocarpan. I assume that the function of OsIRL is unknown and is a homolog of isoflavone reductase, which is involved in the biosynthesis of pterocarpan phytoalexins in legumes.

Authors: Thank you again for pointing out the error. We have now modified this paragraph (line 390-393).

  • Line 509. Is "PAPM" a typo for "PAMP"?

Authors: Sorry for the typo, we have now corrected it in the revised manuscript.

  • Lines 752-753 The citation of Ref. 33 is wrong. Please check the other references carefully again, as I just found this wrong citation by chance.

Zhang, F.; Huang, L.; Zeng, D.; Cruz, C.V.; Li, Z.; Zhou, Y. Comparative proteomic analysis reveals novel insights into the interaction between rice and Xanthomonas oryzae pv. oryzae. BMC Plant Biol. 2020, 20, 563, 10.1186/s12870-020-02769-7.

Authors: Thank you for your carefully checking. We used Endnote for Reference modification. To avoid miss citation, we carefully checked all references again to avoid any possible errors.

Please see the attached manuscript.

Reviewer 3 Report

Manuscript was well designed and written. It will provide very helpful information to the researchers interesting in rice protection from various pathogens. 

Some errors were found in "Reference" setion. Please check them.

Author Response

Thank you for your suggestion, we have checked references again to avoid any error.

Reviewer 4 Report

Abstract:

1 Author may added an efficient information.

2 proteomics, immunity; plant-pathogen interactions may define in abstract by single liner. Introduction: Introduction is not understandable and corelate with each other, can be define more and catchy. 2. Development of proteomics methods and techniques a) under this heading I found information on low to high b) Author never discuss about role of proteomics in Rice and Plant microbe interactions . When author talking about stress it means discuss there abiotic and biotic both stresses . Other than this author can improve and add some recent information related to abiotic stress        

Author Response

1 Author may added an efficient information. proteomics, immunity; plant-pathogen interactions may define in abstract by single liner.

Authors: Thank you for your suggestion. We have now modified the abstract following your suggestion.

Introduction: Introduction is not understandable and corelate with each other, can be define more and catchy. 

Authors: We have now remodified the Introduction part by combining together with proteomics method development part.

  1. Development of proteomics methods and techniquesa) under this heading I found information on low to highb) Author never discuss about role of proteomics in Rice and Plant microbe interactions . When author talking about stress it means discuss there abiotic and biotic both stresses . Other than this author can improve and add some recent information related to abiotic stress  

Authors: Thank you for your comments. We are only focusing on the proteomics investigation of biotic stress response in rice. We therefore remodified this paragraph.

Reviewer 5 Report

In this work, Lirong Wei and collaborators do present an extensive review of what has been published so far to identify proteins involved in plant resistance to microbial pathogens including bacteria, fungi and viruses, using proteomics.

The work is well presented in general and figures represent nicely the main findings.

I would recommend this work for publication once the authors iddress my suggestions and editions.

The overall structure at the beginning of this manuscript is confusing. In my opinion, sections “Introduction” and 2 “Development of proteomics methods and techniques”, should be presented in a single one, especially considering the fact that the authors at the end of point 2 write: “Here we summarize the major proteomic findings with further genetic evidence to illustrate the possible way for future resistant crop designing and application in the field.”

Line 42: “in the Arabidopsis” please delete “the”.

Lines 49-52: “Rice blast disease, caused by the fungal pathogen Magnaporthe oryzae (anamorph Pyricularia oryzae), and bacterial blight disease, caused by the bacterial pathogen Xanthomonas oryzae pv. oryzae, are the most devastating diseases of rice.”

I consider this paragraph should be deleted. The effects of this pathogens are widely discussed in the following sections of this review and it’s confusing to find it here where they are describing the immune responses.

Lines 64-66: this sentence is a little bit confusing. Do the authors mean with “inclusion” that this regulatory elements were introduced in rice from other species? Please clarify.

Line 67: Gumei 4. Maybe researches working in the field are familiar with this, but I would suggest to explain if this is a variety of rice and what is the relevance in this case.

Line 72: “Therefore, even a” please delete “even”.

Line 78: state-of-the-art

Line 83: summarized

Line 95: was an important

Lines 99-101: isobaric tags for relative and absolute quantitation (iTRAQ), tandem-mass tags (TMT), and stable isotope labeling by/with amino acids in cell culture (SILAC) were developed that facilitated even a deeper protein profiling of biological samples.

There are no refences explicitly included.

Lines 103-105: “For instance, the LC-MS/MS-based identification covers almost 70% of protein-coding genes in humans, whereas 104 approximately 50% coverage could be achieved in Arabidopsis [22].” 

This also is confusing. I guess authors mean that 70% of the proteins expressed in humans were identified by this method, not the genes. Moreover, the reference they indicate (number 22) refers to biotic stress in plants, no in humans. Please also add the reference regarding those results in humans.

Line 107: interactions

Figure 1 legend must be improved. For instance: “insect mediator inoculations.”, “protein identification and quantifications”, “The interested protein spots”…

Lines 130-132: “Although the development of protein separation and identification techniques significantly improved the protein detection sensitivity in proteomics approaches, still some technical factors strongly affect the understanding of plant-microbe interactions through the insight view of the proteome.” 

Which are the technical factors that affect this? What the authors mean with “through the insight view”???

Lines 130-147: there are not references at all in this paragraph. Are these statements based on the authors’ previous work? They should include other references supporting these asseverations.

Lines 135-139: “Due to different research focus, the well-infected plant tissues need to be collected at the keynote time point(s), rapidly frozen in liquid nitrogen, and ground into fine powder for protein extraction (Figure 1). Moreover, protein fractionation based on different cell organelles or PTMs in addition to total protein extracts would significantly improve the detection of low abundant proteins or proteins with PTMs (Figure 1).”

Please rewrite: 

Does “different research focus” mean different experimental approaches? 

What the authors want to say by “keynote time points”? 

“protein fractionation based on different cell organelles or PTMs in addition to total protein extracts wouldsignificantly improve the detection of low abundant proteins or proteins with PTMs” Is this an authors’ suggestion or this has been proven in previous published data?

Line 147: Would the authors mean “systematically”? Please check through the whole manuscript.

Line 150: please replace “between” by “with”. 

Line 151: replace “responses” by “respond”.

Line 156: interaction or interactions?

Table 1. In the column labelled as “tissue” strictly speaking they are not talking about the tissue but rather about the sample (leaf total protein, secreted protein and so on). What do they mean by “key proteins”? Proteins in that tissues that are either upregulated or downregulated compared to the control?

Line 172: “were shown to be activated”. Did this researchers measure the activity of these proteins or the authors mean that the protein levels were higher than in the control samples?

Lines 188-189: “Therefore, how prohibitin may involve in the balance of plant defense and growth, however, the molecular mechanism is still elusive.” This sentence must be re-written. 

Lines 197-198: “Since no metabolome results were provided, which strongly limits our understanding of how plant-derived secondary metabolites directly function on bacteria.” Please, re-write. 

Line 202: “JA/ET” please include the full names rather than the abbreviations.

Line 209: “exhibits high resistance to Xoo compared with” Please replace by “exhibts a higher resistance to…”

Line 212: What the authors mean by “alterating”.

Line 225: please replace are by is.

Line 232: “would benefit for investigation” Please re-write.

Line 235-237: this sentence is confusing. Moreover, authors do not indicate what PTM stands for until line 263.

Line 242: “suspension-cultured cells” cell suspension cultures?

Line 245: please replace majorly by mainly in the whole text.

Lines 255-256: indicating the activation of the antioxidant defense system.

Line: 258: “without obvious penalty in yield” meaning?

Line 260: “carbohydrate metabolic process” Do the authors mean carbohydrate metabolism in general or an specific process or pathway?

Figure 2: Line 284 “percepcion” meaning sensing? Most of the labels in this figure are too small, specially the ones inside the “secondary metabolite biosynthesis. I suggest the authors to increase the letter size in this figure.

Line 299: Did the authors mention what Xoc is previously in the text?

Lines 312-314: “therefore triggering the expression of functional proteins involved in bacterial resistance (Figure 2). were rapidly activated/phosphorylated in response to bacterial pathogens” Please re-write.

Line 325: I think this should be another section, from here to line 362.

Line 336: “The host defense response is majorly taken place” Please replace by “The host defense response mainly takes place”.

Line 338: Could the authors explain why the host defense response in the roots and the up-regulation of proteins related to photosynthesis and auxin signaling in the shoots provide evidence that rhizobia endophytes promote rice growth?

Line 342: “was subjected” what authors mean by “subjected”, maybe applied?

Line 346: “metabolism and glycolysis” glysolysis is part of the metabolism, please re-write.

Line 348: may be involved

Line 361: PGPR bacteria?

Line 369: “interaction is switched to a necrotrophic stage”. This is confusing.

Line 377: PTI?

Line 389: please replace “were” by “are”.

Line 395: please replace “is” by “are”.

Lines 402-403: “was regulated by two transcription factorsEthylene-Responsive Element 402 Binding Protein 1 (OsEREBP1) and OsEREBP2, under osmotic stress condition.”

Line 414: “significantly regulated” Do the authors mean that these proteins were either up-regulated or down-regulated, over-expressed or under-expressed…?

Line 439: BTH?

Line 446: please replace “was” by “is”.

Line 478: please replace “were” by “was”.

Line 483: please remove “of”.

Line 497: proteins.

Line 498: it is not clear what authors mean by “were predominant”.

Line 500: “upregulated”. Again, these terms are confusing in the whole text. In my opinion, one thing is protein levels, other up or downregulation, other activation. When authors say upregulated, do they mean that gene expression is enhanced or they are talking about an increase in the protein levels whatever the mechanism responsible for these changes?

Line 527: please replace “for the attenuating” by “for the attenuation”.

Line 530: I’m not sure what the authors mean by “among all alternated”…

Line 535: please replace “process” by “processes”.

Line 539: “was suppressed at the RNA and protein”.

Line 541: “involved”.

Line 544: “alternation” again, is confusing. Perhaps the authors mean “alterations”? Please check in the whole text.

Line 579: “358 proteins differentially regulated proteins”.

Line 586: “could infect” could or it actually infects?

Line 590: “Interestingly, the DUF26 protein was significantly increased”. Please explain what this protein does.

Lines 604-606: please re-write.

Line 538: “suggesting that a balance”.

Line 643: “may be a stratagem”.

Lines 648-649: “whereas M. oryzae triggers ascorbate, and aldarate, and amino sugar and nucleotide sugar biosynthesis-related proteins” could the authors indicate M. ooryzae triggers?.

Line 650: “ABA signaling and in turn enhances”.

Lines 655-657: “In response to the fungal pathogen, proteins related to ET biosynthesis-related protein homocysteine S-methyltransferase (HSM) upon fungal infection”. There is no verb in this sentence…

Line 663: “especially the knockout”.

Line 667: “Therefore, the time-dependent protein dynamic analyses”.

Line 671: “are frequently taken place” please replace by “frequently take place”.

Author Response

The overall structure at the beginning of this manuscript is confusing. In my opinion, sections “Introduction” and 2 “Development of proteomics methods and techniques”, should be presented in a single one, especially considering the fact that the authors at the end of point 2 write: “Here we summarize the major proteomic findings with further genetic evidence to illustrate the possible way for future resistant crop designing and application in the field.”

Authors: Thank you for your suggestion. We have now combined and remodified those two sections following your suggestion.

Line 42: “in the Arabidopsis” please delete “the”.

Authors: We corrected.

Lines 49-52: “Rice blast disease, caused by the fungal pathogen Magnaporthe oryzae (anamorph Pyricularia oryzae), and bacterial blight disease, caused by the bacterial pathogen Xanthomonas oryzae pv. oryzae, are the most devastating diseases of rice.”

I consider this paragraph should be deleted. The effects of this pathogens are widely discussed in the following sections of this review and it’s confusing to find it here where they are describing the immune responses.

Authors: We deleted this sentence.

Lines 64-66: this sentence is a little bit confusing. Do the authors mean with “inclusion” that this regulatory elements were introduced in rice from other species? Please clarify.

Authors: We modified this sentence.

Line 67: Gumei 4. Maybe researches working in the field are familiar with this, but I would suggest to explain if this is a variety of rice and what is the relevance in this case.

Authors: We corrected.

Line 72: “Therefore, even a” please delete “even”.

Authors: We corrected.

Line 78: state-of-the-art

Authors: We corrected.

Line 83: summarized

Authors: We corrected.

Line 95: was an important

Authors: We corrected.

Lines 99-101: isobaric tags for relative and absolute quantitation (iTRAQ), tandem-mass tags (TMT), and stable isotope labeling by/with amino acids in cell culture (SILAC) were developed that facilitated even a deeper protein profiling of biological samples.

There are no refences explicitly included.

Authors: We added reference in the paragraph.

Lines 103-105: “For instance, the LC-MS/MS-based identification covers almost 70% of protein-coding genes in humans, whereas 104 approximately 50% coverage could be achieved in Arabidopsis [22].” 

This also is confusing. I guess authors mean that 70% of the proteins expressed in humans were identified by this method, not the genes. Moreover, the reference they indicate (number 22) refers to biotic stress in plants, no in humans. Please also add the reference regarding those results in humans.

Authors: We have now modified this sentence to improve the readability and added required references.

Line 107: interactions

Authors: We corrected.

Figure 1 legend must be improved. For instance: “insect mediator inoculations.”, “protein identification and quantifications”, “The interested protein spots”…

Authors: We modified the figure legend.

Lines 130-132: “Although the development of protein separation and identification techniques significantly improved the protein detection sensitivity in proteomics approaches, still some technical factors strongly affect the understanding of plant-microbe interactions through the insight view of the proteome.” 

Which are the technical factors that affect this? What the authors mean with “through the insight view”???

Authors: We modified the sentence.

Lines 130-147: there are not references at all in this paragraph. Are these statements based on the authors’ previous work? They should include other references supporting these asseverations.

Authors: We modified the sentence and added references.

Lines 135-139: “Due to different research focus, the well-infected plant tissues need to be collected at the keynote time point(s), rapidly frozen in liquid nitrogen, and ground into fine powder for protein extraction (Figure 1). Moreover, protein fractionation based on different cell organelles or PTMs in addition to total protein extracts would significantly improve the detection of low abundant proteins or proteins with PTMs (Figure 1).”

Please rewrite: 

Does “different research focus” mean different experimental approaches? 

What the authors want to say by “keynote time points”? 

“protein fractionation based on different cell organelles or PTMs in addition to total protein extracts wouldsignificantly improve the detection of low abundant proteins or proteins with PTMs” Is this an authors’ suggestion or this has been proven in previous published data?

Authors: We modified the sentence and added references.

Line 147: Would the authors mean “systematically”? Please check through the whole manuscript.

Authors: We mean a systematic biology approach. We modified the sentence and check through the manuscript.

Line 150: please replace “between” by “with”. 

Authors: We corrected.

Line 151: replace “responses” by “respond”.

Authors: We corrected.

Line 156: interaction or interactions?

Authors: We corrected.

Table 1. In the column labelled as “tissue” strictly speaking they are not talking about the tissue but rather about the sample (leaf total protein, secreted protein and so on). What do they mean by “key proteins”? Proteins in that tissues that are either upregulated or downregulated compared to the control?

Authors: We changed “tissue” to “sample”, and replaced “key proteins” to “differentially accumulated proteins”.

Line 172: “were shown to be activated”. Did this researchers measure the activity of these proteins or the authors mean that the protein levels were higher than in the control samples?

Authors: We corrected.

Lines 188-189: “Therefore, how prohibitin may involve in the balance of plant defense and growth, however, the molecular mechanism is still elusive.” This sentence must be re-written. 

Authors: We rewrote the sentence.

Lines 197-198: “Since no metabolome results were provided, which strongly limits our understanding of how plant-derived secondary metabolites directly function on bacteria.” Please, re-write. 

Authors: We removed this sentence.

Line 202: “JA/ET” please include the full names rather than the abbreviations.

Authors: We added full name of JA and ET.

Line 209: “exhibits high resistance to Xoo compared with” Please replace by “exhibts a higher resistance to…”

Authors: We corrected.

Line 212: What the authors mean by “alterating”.

Authors: We corrected.

Line 225: please replace are by is.

Authors: We corrected.

Line 232: “would benefit for investigation” Please re-write.

Authors: We rewrote the sentence.

Line 235-237: this sentence is confusing. Moreover, authors do not indicate what PTM stands for until line 263.

Authors: We rewrote the sentence.

Line 242: “suspension-cultured cells” cell suspension cultures?

Authors: We corrected.

Line 245: please replace majorly by mainly in the whole text.

Authors: We checked and corrected.

Lines 255-256: indicating the activation of the antioxidant defense system.

Authors: We corrected.

Line: 258: “without obvious penalty in yield” meaning?

Authors: We modified this sentence.

Line 260: “carbohydrate metabolic process” Do the authors mean carbohydrate metabolism in general or an specific process or pathway?

Authors: We corrected.

Figure 2: Line 284 “percepcion” meaning sensing? Most of the labels in this figure are too small, specially the ones inside the “secondary metabolite biosynthesis. I suggest the authors to increase the letter size in this figure.

Authors: We remodifed the figure and increased letter size.

Line 299: Did the authors mention what Xoc is previously in the text?

Authors: It was mentioned in previous paragraph (Line 189-190).

Lines 312-314: “therefore triggering the expression of functional proteins involved in bacterial resistance (Figure 2). were rapidly activated/phosphorylated in response to bacterial pathogens” Please re-write.

Authors: We rewrote this sentence.

Line 325: I think this should be another section, from here to line 362.

Authors: We sperated this part to be a new section.

Line 336: “The host defense response is majorly taken place” Please replace by “The host defense response mainly takes place”.

Authors: We corrected.

Line 338: Could the authors explain why the host defense response in the roots and the up-regulation of proteins related to photosynthesis and auxin signaling in the shoots provide evidence that rhizobia endophytes promote rice growth?

Authors: We are sorry for overstating. We modified this sentence.

Line 342: “was subjected” what authors mean by “subjected”, maybe applied?

Authors: We corrected.

Line 346: “metabolism and glycolysis” glysolysis is part of the metabolism, please re-write.

Authors: We rewrote this sentence.

Line 348: may be involved

Authors: We corrected.

Line 361: PGPR bacteria?

Authors: We corrected.

Line 369: “interaction is switched to a necrotrophic stage”. This is confusing.

Authors: We corrected.

Line 377: PTI?

Authors: We corrected.

Line 389: please replace “were” by “are”.

Authors: We corrected.

Line 395: please replace “is” by “are”.

Authors: We corrected.

Lines 402-403: “was regulated by two transcription factorsEthylene-Responsive Element 402 Binding Protein 1 (OsEREBP1) and OsEREBP2, under osmotic stress condition.”

Authors: We modified this sentence.

Line 414: “significantly regulated” Do the authors mean that these proteins were either up-regulated or down-regulated, over-expressed or under-expressed…?

Authors: We corrected.

Line 439: BTH?

Authors: We added information.

Line 446: please replace “was” by “is”.

Authors: We corrected.

Line 478: please replace “were” by “was”.

Authors: We corrected.

Line 483: please remove “of”.

Authors: We corrected.

Line 497: proteins.

Authors: We corrected.

Line 498: it is not clear what authors mean by “were predominant”.

Authors: We corrected.

Line 500: “upregulated”. Again, these terms are confusing in the whole text. In my opinion, one thing is protein levels, other up or downregulation, other activation. When authors say upregulated, do they mean that gene expression is enhanced or they are talking about an increase in the protein levels whatever the mechanism responsible for these changes?

Authors: We corrected and checked through the manuscript.

Line 527: please replace “for the attenuating” by “for the attenuation”.

Authors: We corrected.

Line 530: I’m not sure what the authors mean by “among all alternated”…

Authors: We corrected.

Line 535: please replace “process” by “processes”.

 Authors: We corrected.

Line 539: “was suppressed at the RNA and protein”.

Authors: We corrected.

Line 541: “involved”.

Authors: We corrected.

Line 544: “alternation” again, is confusing. Perhaps the authors mean “alterations”? Please check in the whole text.

Authors: We corrected.

Line 579: “358 proteins differentially regulated proteins”.

Authors: We corrected.

Line 586: “could infect” could or it actually infects?

Authors: We corrected.

Line 590: “Interestingly, the DUF26 protein was significantly increased”. Please explain what this protein does.

Authors: We modified this sentence.

Lines 604-606: please re-write.

Authors: We rewrote this sentence.

Line 538: “suggesting that a balance”.

Authors: We corrected.

Line 643: “may be a stratagem”.

Authors: We corrected.

Lines 648-649: “whereas M. oryzae triggers ascorbate, and aldarate, and amino sugar and nucleotide sugar biosynthesis-related proteins” could the authors indicate M. ooryzae triggers?.

Authors: We rewrote this sentence.

Line 650: “ABA signaling and in turn enhances”.

Authors: We corrected.

Lines 655-657: “In response to the fungal pathogen, proteins related to ET biosynthesis-related protein homocysteine S-methyltransferase (HSM) upon fungal infection”. There is no verb in this sentence…

Authors: We rewrote this sentence.

Line 663: “especially the knockout”.

Authors: We corrected.

Line 667: “Therefore, the time-dependent protein dynamic analyses”.

Authors: We corrected.

Line 671: “are frequently taken place” please replace by “frequently take place”.

Authors: We corrected.

Reviewer 6 Report

Minor remarks:

line 57 - ", whereas"

line 151 - how plants respond

Author Response

Thank you for your comments. We have modified the manuscript as suggested.

line 57 - ", whereas"

Authors: We corrected.

line 151 - how plants respond

Authors: We corrected.